# Path toward Sustainability in Wastewater Management in Brazil

**DOI:** 10.3390/ijerph20166597

**Published:** 2023-08-18

**Authors:** Débora Cynamon Kligerman, Aline Stelling Zanatta, Graziella de Araújo Toledo, Joseli Maria da Rocha Nogueira

**Affiliations:** 1Departamento de Saneamento e Saúde Ambiental (DSSA) (ENSP), Fundação Oswaldo Cruz, FIOCRUZ, Rio de Janeiro 21041-210, Brazil; graziellatoledo@gmail.com; 2Instituto de Ciência e Tecnologia em Biomodelos (ICTB), Fundação Oswaldo Cruz, FIOCRUZ, Rio de Janeiro 21040-360, Brazil; aline.zanatta@fiocruz.br; 3Laboratório de Microbiologia do Departamento de Ciências Biológicas (DCB) (ENSP), Fundação Oswaldo Cruz, FIOCRUZ, Rio de Janeiro 21041-210, Brazil; joseli.maria@fiocruz.br

**Keywords:** wastewater management, sustainability, public policy, Brazil

## Abstract

Developing countries have not carried out the adequate management of wastewater and are a long way off meeting the sustainability goal of universal access to safely managed sanitation services by 2030. This article discusses sustainability in wastewater management and conducts a narrative literature review to analyze four stages on the path toward sustainability: (1) the prevention of or reduction in pollution at the source; (2) wastewater collection and treatment; (3) using wastewater as an alternative source of water; and (4) the recovery of useful by-products. It also provides an overview of wastewater management in Brazil and shows the advantages of using wastewater to produce biofuel in a country in which 48.3% of energy production comes from renewable sources.

## 1. Introduction

One of the key challenges facing developing countries is to ensure access to adequate sanitation. In 2020, 494 million people still practiced open defecation [1]. Since 2015, UN Member States have adopted the 2030 Agenda for Sustainability, consisting of 17 Sustainability Goals (SDGs) and 169 associated targets that should be achieved by 2030. Goal 6 is “Ensure availability and sustainable management of water and sanitation for all” [2].

The UN report Shared Responsibility, Global Solidarity: Responding to the socio-economic impacts of COVID-19 [3,4] analyzes the negative effect of the pandemic on sustainability efforts. In Brazil, investment toward improving access to sanitation has been outweighed by the pace of population growth and the formation of precarious human settlements [5].

According to the United Nations World Water Development Report [6], sustainable wastewater management can be broken down into the following stages or phases: (1) the prevention of or reduction in pollution at the source; (2) wastewater collection and treatment; (3) using wastewater as an alternative source of water; and (4) the recovery of useful by-products. Developed countries are already in phases 3 and 4, while less developed countries still find themselves in phases 1 and 2. Despite its relatively high Gross Domestic Product (GDP), which is the sum of all goods and services produced by a country, Brazil has yet to achieve universal access to basic sanitation and is in between phases 1 and 2.

This article discusses sustainability in wastewater management and the use of wastewater as a raw material for energy production as a path toward sustainability in Brazil.

We conducted a narrative literature review of documents and articles published over the past 10 years using the keywords “wastewater management”, “sustainability”, and “public policies”. In addition, we used reports published by the Ministry of Health [6], the UN (2017) [2], and UNICEF & WHO (2017 [7]; 2021 [8]), which are key reference publications on this topic and contain important proposals for government programs that address the sustainability of wastewater management globally.

We begin by discussing the importance of adequate sanitation for sustainability and then provide an overview of sanitation in Brazil. We then go on to discuss the use of treated wastewater as an alternative to using potable water and describe some added-value by-products of sewage. Finally, we look at the prospects for renewable energy production using wastewater in Brazil.

## 2. Sustainability and Sanitation

Sustainability is a concept that involves the rational use of our natural resources so as not to compromise the use of these resources by future generations [9].

SDG 6 includes ending open defecation (6.2) and improving water quality by reducing pollution and halving the proportion of untreated wastewater (6.3) [2], reinforcing the interfaces between universal access to adequate sanitation, improving water quality, efficient use of water resources, and strengthening integrated water resources management [4]. It is important to clarify that the concept of sanitation used by the UN encompasses access to infrastructure, facilities, and services for the management and disposal of solid and liquid waste [5].

In 2017, baselines were set for the above targets in the form of “service ladders” corresponding to the status of services. Table 1 shows the service levels for sanitation services. To meet the minimum standard for sanitation services (basic), the family should have a bathroom within the dwelling, yard, or plot. To meet the highest standard for sanitation services (safe management), the family should have a bathroom on the premises where excreta are flushed out of the household (to a septic tank) or collected and transported for treatment in adequate facilities (such as a wastewater treatment plant) [7].

Between 2015 and 2020, efforts were stepped up to improve the level of sanitation services [8]. The population practicing open defecation decreased from 10% to 6% (494 million people in 55 countries), while the percentage of the global population using unimproved facilities decreased from 10% to 8% (616 million people). Around 7% of the population had limited services in 2020 (580 million people) and the percentage of people with basic services decreased from 26% to 24% (1.9 billion people). The proportion of the population using safely managed services increased from 47% to 54% (44% in rural areas and 62% in urban areas). However, 3.6 billion people still lacked safely managed services. To meet SDG 6 (6.2)—universal access to safely managed sanitation services—there will need to be a 4-fold increase in the overall rate of progress and 15-fold increase in the rate in less developed countries (Figure 1).

In 2020, 120 countries had estimates for safely managed sanitation services. In North America, Europe, Australia, and New Zealand, coverage ranged from 75% to 99%. In Latin America, only Chile had achieved higher levels of coverage (between 75% and 99%). Percentages were below 25% in Bolivia, Venezuela, and Colombia, while in Brazil, between 25% and 50% of the population had access to safely managed services [8] (Figure 1).

In contrast, in 55 countries, at least 5% of the population still practiced open defecation in 2021. In North America, Europe, Australia, and New Zealand, the proportion was less than 1%. In Argentina, Brazil, Chile, Paraguay, and Uruguay, less than 1% of the population practiced open defecation, while in Peru, Colombia, and Venezuela, rates varied from 1% to 5%. In Bolivia, rates varied from 5% to 25% [8].

To achieve sustainability, an in-depth analysis is required to ascertain which stage of wastewater management a country is at. The four stages outlined above generate social, economic, and environmental benefits and contribute to water security and sustainability.

The first stage, the reduction in pollution at the source, includes prohibiting or controlling the use of certain contaminants to limit their entering into wastewater streams. Many countries with regulatory instruments and enforcement processes in place have completed this stage. Examples include the United States—with the Clean Water Act (1948) [10] and pollution control instruments Total Maximum Daily Load (TMDL), National Pollutant Discharge Elimination System (NPDES), and the Assessment, Total Maximum Daily Load Tracking and Implementation System (ATTAINS)—and Europe—with the National Environment (Standards for Discharge of Effluent into Water or on Land) Regulations, S.I. No. 5 of 1999 [11]. However, this does not necessarily guarantee that regulatory instruments are properly implemented or that management is adequate.

The second stage brings together countries that already have sewerage systems and wastewater treatment plants (WWTPs). It is estimated that 60% of the global population is connected to sewerage systems; however, only a small proportion of collected wastewater is treated. Developed countries treat around 70% of their wastewater, compared to between 28% and 38% in developing countries. In contrast, in the world’s poorest countries, only 8% of wastewater is treated [6]. Most countries that treat their wastewater use centralized WWTPs; however, decentralized wastewater treatment systems are often more feasible for individual or small groups of properties as they are between 20% and 50% cheaper than conventional systems and maintenance costs are between 5% and 25% lower. Furthermore, decentralized systems promote nutrient and energy recovery, water reuse, and minimize water shortages in areas where water is scarce [6].

In the third stage, wastewater is used as an alternative source of water. While untreated or diluted wastewater has been used for centuries for irrigation, the reuse of treated wastewater is more recent and presents an opportunity for both industry and municipal water utilities. The feasibility of sewage reuse depends on the characteristics of the sewage, level of treatment, and quality of the final water and for which use it is intended. Reuse is intended to reduce potable water use and promote nutrient recovery and aquifer recharge. This type of approach, in addition to contributing to cost reduction, also meets the precautionary principle [6]. However, one has to consider the health risk of contracting infections and diseases when reusing treated sewage. This risk depends on the amount and dispersion of bacteria in the treated sewage. For use in irrigation, there may be a risk of contracting intestinal parasites from consuming vegetables irrigated with treated sewage. Also, sewage can contain antibiotics and hormones, which can cause the proliferation of resistant bacteria and inhibit the biological treatment process, and hormones can cause risks to human health [12]. Also, related to financial analysis, the revenue from the sale of treated wastewater does not cover the cost of water treatment [6]. Furthermore, governmental subsidies would be misapplied in this case, as they do not reach the low-income population, which is not connected to the grid [12].

The fourth stage, the exploration of wastewater by-products, such as energy and nutrients, is still in the early stages of development. Energy can be obtained through sludge, which generates biogas or electricity. This process contributes to the reduction in greenhouse gases, it generates carbon credits, and thus reduces a WWTP’s operating costs. Nutrients, such as nitrogen and phosphorus, can be obtained through the separation of urine from feces. It is important to highlight that mineral phosphorus is scarce and may even become exhausted in the coming decades, making its recovery from wastewater a desirable and viable alternative [6].

As mentioned above, the level of sustainability in wastewater management varies between developed and developing countries. Developed countries are already at the reuse and recovery of wastewater by-products stages, while wastewater treatment levels are generally low in developing countries and sewage is often discharged into natural water courses, hampering the ability of these countries to comply with SDG 6.3 (reducing water pollution). To achieve sustainability, it is also necessary to increase social acceptance of wastewater use and raise awareness of the economic benefits of by-products, which offer business opportunities from the recovery of energy, nutrients, and metals [6].

## 3. Overview of Sanitation in Brazil

Brazil is marked by poor access to sanitation services, which includes both wastewater collection and treatment. According to data from the National Sanitation Information System (SNIS) published in 2022 [13], the country collected an average of 55.8% of the wastewater produced in the country (Figure 2), of which only 51.2% was treated.

The 2017 National Basic Sanitation Survey (PNSB) included a study of the level of treatment practiced in treatment plants in operation in the country and the most efficient alternatives. Of the 3206 municipalities with wastewater collection services (57.5% of the total number of municipalities in the country), 2013 (36.1%) treated their sewage. The regions with the largest number of municipalities with WWTPs in operation were the Midwest and South (94.4% and 71.7%, respectively), while the region with the smallest number was the Northeast (51.2%) [14]. Regarding the population with access to wastewater collection, there are huge differences between the Brazilian regions. In the Southeast region, 80.5% of the population has wastewater collection, while in the North region, only 13.1% of the population has this service. In the Midwest, South, and Northeast regions, 59.46%, 47.37%, and 30.29%, respectively, of the population has wastewater collection [15] Meanwhile, the deficit in wastewater collection by income range is 44.9% for the population earning up to half a minimum wage, 27.9% for the population with an income range from half to one minimum wage, 39.3% for the population with an income range from one to two minimum wages, 4.9% for an income range from two to three minimum wages, and only 4.1% for an income range greater than three minimum wages [15]. The minimum wage in May 2023 was $276.73.

There are four stages in the wastewater treatment process: preliminary, primary, secondary, and tertiary. Preliminary treatment consists of the removal of coarse solids using screens and desanders, which remove grit. Grease and oils are also removed during this stage in grease chambers. Primary treatment removes settleable organic and inorganic solids. During secondary (biological) treatment, the organic loading of the effluent is reduced through oxidation performed by microorganisms. This is the most common type of treatment in WWTPs in Brazil. The last stage is tertiary treatment (physical–chemical process), which removes dissolved and suspended inorganic solids, pathogens, nitrogen, phosphorous, and other non-biodegradable compounds [16].

In 2020, 1470 (26.4%) of the 2013 municipalities that treated their sewage performed secondary treatment [14].

The PNSB also addressed the question of reuse. Only 3.6% of the WWTPs used the treated effluent. The sludge generated during the treatment process received treatment in 50.1% of the municipalities with WWTPs in operation. In the Midwest, Southeast, North, and Northeast, 13.9% of municipalities disposed of the treated or untreated sludge in landfills. In the South, 70.7% of municipalities treated sludge and the main destination of this waste was agricultural purposes [13].

There is an urgent need, both from an environmental and public health point of view, to increase the proportion of treated wastewater, as improvements in service provision bring visible medium- to long-term benefits. Domestic and industrial wastewater is one of the leading sources of contamination and pollution of water bodies and groundwater [6]. It is estimated that the subsoil receives a huge amount (4329 Mm^3^ per year) of wastewater due to the lack of collection and this situation is worse in poor areas with a higher population density, in areas with older sewage pipes, and in areas that use pits [17]. The lack of investment in wastewater collection and treatment infrastructure directly compromises the quality of distributed water and, above all, severely affects the quality and efficiency of the country’s sanitation services [16].

According to a study undertaken in 2013 for the National Basic Sanitation Plan (PLANSAB) [18], $1.90 billion would have to be spent annually between 2014 and 2033 to achieve universal access to adequate sanitation services in Brazil. Brazil has witnessed a decrease in investment in sanitation in real terms, with spending remaining relatively stable over recent years: 2014, $1.15 billion; 2015, $1.08 billion; 2016, $0.87 billion; 2017, $0.80 billion; 2018, $0.98 billion; and 2019, $1.10 billion [19,20,21,22,23,24].

Funding shortfalls affecting the sanitation infrastructure and services result in economic losses in other areas. In 2015, around 98 million people did not have access to wastewater collection services, leading to significant health costs (hospital admissions and medication) associated mainly with the high incidence of diarrheal disease. The benefits of improving access to wastewater collection and treatment services include improvements in the quality of life, which in turn lead to an increase in productivity, increased property prices, and a reduction in diseases transmitted by vectors who lay their eggs in water, such as the mosquito *Aedes Aegypti*, which transmits dengue, Zika, yellow fever, and Chikungunya [25].

Since a large part of the wastewater produced in the country ends up in rivers, lakes, and the sea, poor sanitation and wastewater management contributes to the further degradation of water bodies, with a range of negative consequences for the environment. Brazil has legislation that establishes standards for the discharge of wastewater into receiving waters, as well as for the classification of water bodies and the assessment of bathing water quality, including Conama (National Environment Council) resolutions 274/00 [26], 357/05 [27], and 430/11 [28].

## 4. Use of Treated Wastewater as an Alternative Source of Water

Water is essential for life on Earth and humanity has always regarded it to be an infinite resource; however, clean drinking water is increasingly hard to find. Studies across the globe have shown that water scarcity combined with poor water quality will be one of the most pressing challenges faced by humanity this century, affecting various countries [29,30].

Water reuse is being proposed as an alternative to drinking water, but its implementation requires everything from regulatory measures to better management performance and depends on innovative technologies for transforming wastewater into drinking water [6].

Water reuse is the process of converting wastewater into water that can be reused for other purposes [30,31]. This process has always existed in nature and performing it artificially has now become an interesting and necessary option [31,32].

The treatment of wastewater for reuse, accelerating natural processes that occur in the environment to overcome water scarcity in the context of a circular economy, is critical for maintaining healthy conditions for the survival of life on our planet over the coming decades [33,34].

Growing water demand, coupled with the deterioration of infrastructure, poorly directed investment, and overexploited water sources, threaten water security around the world [30]. Reclaimed water, as treated wastewater is called, can significantly reduce the consumption of freshwater in activities that do not need perfectly clean drinkable water. When the source of reclaimed water (WWTP) is close to where it will be used, costs are even lower. Considering that the type of treatment used to convert wastewater into water for reuse can be tailored to each purpose, this cost can fall even further [6].

However, the reuse of water can be a very complex task and requires well-thought-out solutions adapted to local circumstances and needs. In industrial parks, it can be used for heating/cooling systems [35]. In rural areas, wastewater can be used for irrigation, while in residential areas, graywater can be used for toilet flushing, watering the garden, parks, and yards, and washing streets. In the slums area, however, graywater is directly discharged into the ground, and becomes dangerous, because graywater usually contains detergents, soap, and shampoos that contain xenobiotic organic compounds, which are endocrine disruptors and can impact soil and plants [36].

Brazil does not have standardized updated norms and regulations governing the types of reuses of water. The only national norms and regulations are those issued by the Brazilian Technical Standards Association (ABNT) in 1997 and the most up-to-date norms in the country are those published by the Government of the State of Ceará in 2016 [37].

Moura et al. (2020) [38] proposed an updated classification considering the origin and end uses of wastewater: water for local or internal reuse, so-called graywater, which can be used in the home; water for external reuse, so-called blackwater, generated from the toilet and that should be treated in a WWTP; and water for industrial reuse from industrial activities that, depending on each case, can be reused directly internally or externally.

It is important to stress that the recommended treatment, safety criteria, and capital, operating, and maintenance costs will depend on the quality of the wastewater and specific end use [39].

The reuse of water in Brazil faces complex challenges. A leading example is differences in water availability across regions, with some regions having high levels of natural water availability and others, such as the semi-arid region, experiencing extreme scarcity [40].

Another aspect that needs to be addressed is public awareness regarding the rational use of water. The fact that Brazil has the highest volume of freshwater resources in the world has generated the false idea among the population that water is an inexhaustible resource. This hampers efforts to raise awareness of the finiteness of water and the need for alternatives such as reclaimed water, which is perfectly suitable for activities that do not require drinking water [40].

It is also critical to raise awareness among the political class of the feasibility of water reuse and the fact that water recycling can become a widespread reality in the future [32]. The reuse of water in Brazil therefore needs to be promoted, institutionalized, and regulated to ensure that it is practiced in accordance with adequate technical standards, economically feasible, socially acceptable, and safe, both in terms of health and the environment [38,39].

However, regulation is a reality in only eight states in Brazil: Bahia, Ceará, Espirito Santo, Minas Gerais, Paraná, Rio Grande do Sul (in Caxias do Sul), Rio de Janeiro, and São Paulo [38].

Finally, recycled water is the only resource that grows with increasing economic growth. It is estimated that, today, 380 billion cubic meters of wastewater are produced annually worldwide. This amount is expected to reach 470 billion by 2030 and 574 billion by 2050 [33,34].

In addition to the economic benefits, the use of reclaimed water generates benefits for ecosystems because it reduces freshwater abstraction [6] and minimizes the risk of surface and groundwater pollution. The situation worsens in slums where there are no sewage systems and sewage and graywater are dumped on the ground in open ditches [41]. However, a study carried out in a slum in Uganda carried out pretreatment followed by double filtration using coarse and fine media and managed to reduce graywater pollution by 60% [42], indicating that with political will and technology, a solution to this problem can be achieved. It is also a critical part of the circular economy and strengthens water self-sufficiency, giving countries access to a reliable local water resource protected from neighboring countries with ineffective environmental policies [43].

## 5. Use of Wastewater By-Products

Wastewater management models have changed over the years. The focus became the recovery of resources, not just the disposal of waste directly into the rivers or seas. Population growth, combined with an increase in the standard of living, has become a major challenge in terms of waste management. The increase in consumption requires waste prevention, recycling, and detoxification, through an effective system of separation and conversion of organic contaminants and heavy metals into non-toxic substances [44].

When it comes to wastewater, this means: (a) separating and neutralizing toxic compounds such as microplastics, pharmaceuticals, surfactants, and heavy metals; (b) recovering nutrients such as nitrogen and phosphorous for use as agricultural fertilizers using different techniques, replacing the use of these scarce minerals; and (c) recovering the energy content of carbon compounds by bio-fermentation, generating biogas [44].

It should be noted that the change in the lifestyle of populations has caused the presence of contaminants in wastewater to increase. Such substances, during the treatment of wastewater, accumulate in the sewage sludge and, during the stabilization of the waste, some cannot be decomposed [45,46].

It is thus essential to consider toxicity when defining the end use of sludge [47,48] and monitor the presence of hazardous substances in waste. In addition, it is important to investigate possible treatment methods and the sequence of processes needed to break down toxic substances. Norms and regulations should be respected. It is also important to carry out a cost–benefit analysis to assess the viability of these methods.

The increase in consumption requires waste prevention, recycling, and detoxification, through an effective system of separation and conversion of organic contaminants and heavy metals into non-toxic substances [49]. The use of these two by-products is detailed in the following items.

### 5.1. Biogas

Biogas is a clean and renewable energy, based on a well-established technology, in which microorganisms carry out the anaerobic digestion of organic waste, following several metabolic pathways to decompose organic matter, creating added value during the decomposition of organic matter [50]. This resource is one of the most used sludge by-products for energy generation. The energy contained in wastewater can be used for space heating and cooling or electricity generation. WWTPs therefore have the potential to transition from being energy consumers to producers, significantly reducing their carbon footprint [6]. The benefits of the recovery of energy from wastewater in the form of biogas for the environment are therefore twofold, providing a way to minimize emissions of greenhouse gases (GHGs) such as methane and a source of renewable energy [6].

Biogas is produced by anaerobic digestion, a process in which microorganisms break down organic matter, such as manure, municipal solid waste, sewage sludge, and biodegradable and agricultural waste, in an oxygen-free environment, making it an important source of sustainable clean energy for the generation of heat and electricity [51].

This resource is currently considered a promising renewable energy source. Biogas has in its composition about 60% methane, but this index depends on the composition of the sewage sludge [51].

The anaerobic digestion process typically takes around 30 days, with 80–85% of the biogas being produced in the first 15–18 days. Production rates are higher at a temperature range of 30–60 °C and pH of between 5.5 and 8.5. The estimated annual production capacity of a biogas plant with a 500 m^3^ digestion chamber is 20–36 × 10^3^ Nm^3^.

Calculations also show that 1 m^3^ of biogas with 60% methane contains the equivalent of 6–6.5 kWh of energy, with average residential energy consumption being 5.2 kWh per day [52,53].

A study in a municipality in Recife, in the state of Pernambuco, showed that recovered biogas from waste produced in the town had the potential to generate more than 265 MJ of thermal energy a day, which is equivalent to over 486,000 kWh of electric energy, enough to supply 93,031 households [52].

Biogas is therefore one of the most important sources of renewable and sustainable energy, which in turn are the best substitute for conventional fuels and energy sources. Apart from generating renewable energy without harming the environment, the production of biogas from sewage sludge is resource efficient, because it uses relatively cheap and simple reactors and operational procedures [51].

There are different ways of classifying the operations of biogas plants depending on the effluent, temperature, and shape of the reactor. According to Borowski et al. (2014) [54] and Cabbai et al. (2013) [55], an evaluation carried out using biochemical methane potential tests showed that sewage sludge can provide a methane yield of 249–274 mL-CH4/g VS.

Advances in anaerobic digestion technology have led to a significant reduction in production time and costs, enabling a rapid progression in the understanding of this complex process [50], including not only microbial composition and syntrophic interactions within the microbiome, but also the expression of different genes under different environmental conditions, which, through the use of bioinformatics, can increase the yield of anaerobic digestion, generating enormous potential that can be explored in the future. Metagenomics, metatranscriptomics, metaproteomics, metabolomics, or stable isotope probing are used to facilitate specific metabolic processes with microbial species [56,57], improving the efficiency of the digestion process and increasing the amount of biomass that can be treated. According to Kougias and Angelidaki (2018) [50], there is no doubt that microbiology will play a key role in the diagnosis and monitoring of the anaerobic digestion process through the exploration of specific biomarkers.

Based on these studies, it is believed that advanced monitoring and control will play an essential role in future biogas plants, significantly contributing to process optimization. Ongoing monitoring of biogas composition should be performed to detect the presence of high hydrogen sulfide concentrations and remove excess content [58].

Regarding the cost–benefit analysis of biogas production, it is important to highlight that the minimum cost of the electricity generated from biogas from sewage sludge was higher than the average price of electricity in Brazil [59]. Currently, therefore, biogas is not an economically attractive resource for household electricity generation in a country like Brazil. However, recovered biogas can reduce WWTP energy costs [60] and has the potential to increase their operational efficiency [61]. One study in Italy analyzed 202 WWTPs comparing the energy used in aerobic and anaerobic treatment and concluded that while aerobic treatment on average spends 1.02 kWh/m^3^, anaerobic treatment only spends 0.43 kWh/m^3^. Around the world, the energy consumption from anaerobic digestion was between 0.4 and 0.7 kWh/m^3^. In terms of population equivalent (PE), the energy consumed by anaerobic treatment was 24.4 kWh/PE per year and with aerobic treatment it was much higher, 50.9 kWh/PE per year [62].

Amaral et al. (2019) [63] show that biogas generated in WWTPs can be used to dry sludge used as a fertilizer in agriculture, which has a high overall sustainability rating in the Dashboard of Sustainability (DoS).

In addition to the conversion of CO_2_ to biomethane, it is believed that biogas can be used to produce molecules with multiple potential uses. Methane gas, for example, can be used as a source of carbon and energy for some specific microorganisms, such as methanotrophs, which are responsible for gas consumption in terrestrial and aquatic environments, in addition to having potential to produce polyhydroxyalkanoates [50,64].

It is expected that biogas will be exploited and that it can also be used for the production of high added-value products, such as proteins, extracellular polysaccharides, bioplastics, and chemical products, such as biosuccinic acid, hexanol, and lactic acid [50]. The implementation of these processes on an industrial scale faces a number of biotechnological challenges, which can be successfully addressed in Brazil by interdisciplinary research. However, these findings are in the experimental phase and Brazil and Latin America need solutions that are implemented in the short-term, with feasible and reliable technology.

In Feira de Santana, in the state of Bahia, a project developed to recover biogas from sewage sludge to reduce the energy costs of a WWTP (Jacuípe II) resulted in considerable cost savings, with the monthly electricity bill decreasing from $5358 to $1030. In addition to reducing energy bills, which is one of the biggest outgoings of WWTPs, another benefit of recovering biogas from sewage sludge is the significant reduction in GHG emissions [65]. In another study conducted in Brazil, it was observed that methane production was 1427.2 m^3^/d and the average electrical energy available was 65,280.3 kWh/month, which represents 59% of the average monthly demand of WWTPs [66]. So, this contributes to a reduction in the electricity cost [67]. It should be clarified that the energy consumption in a WWTP varies from 0.26 to 1.6 kWh/m^3^. It also can be measured as being from 12 to 60 kWh/population equivalent per year. Considering countries, the production of biogas from sewage sludge in Brazil is 42 Gwh per year, while in Germany, it is 3050 Gwh. [67].

### 5.2. Agricultural Use of Sewage Sludge

The recovery of nutrients for agriculture can add significant new value streams [6]: 13.4% of global agricultural demand for nitrogen, phosphorous, and potassium can be recovered from sewage. However, to harness this potential, it is necessary to improve the efficiency of nutrient-recovery processes [33].

The most common method of harnessing nutrients in wastewater is the use of sewage sludge as fertilizer, in response to the increase in the production of sludge and growing demand for fertilizer from agriculture. Anaerobically digested sewage sludge is a valuable source of organic material and nutrients and significant advances have been made in recent years in nutrient-recovery technologies. Plants benefit not only from the water, but also the compounds dissolved in domestic sewage, such as organic material, nitrogen, phosphorous, potassium, and micronutrients [68]. Sewage sludge can be used through landspreading, where it is simply applied to the soil. This method improves the soil structure and water infiltration and adds nutrients like nitrogen and potassium. However, the direct application of sludge to land should comply with the limits on the concentration of heavy metals, nitrates, and other pollutants. In addition, sewage sludge can also be used in cement and brick production. One of the disadvantages of landspreading is a lack of information on the concentration of nutrients and bioavailability, making sludge a low-quality fertilizer in comparison with conventional fertilizers. In contrast to direct application, with pyrolysis, most of the phosphorous and half of the nitrogen is retained in a charcoal-like sludge char, which is a potential fertilizer. While the nitrogen in sludge char is insoluble, this by-product has a high available phosphorous content [68].

Although the amount of excreted nutrients depends on the quantity and quality of the food consumed, the fact that urine is rich in nitrogen and feces are always rich in phosphorus, potassium, and organic matter makes it possible to use these excreta as fertilizers in agriculture, since in addition to being able to function as high-quality fertilizers, they also release low levels of contaminants such as heavy metals.

Phosphorous and nitrogen in the soil are easily absorbed by plants [69] but are not readily bioavailable in sewage sludge generated in WWTPs, meaning it is important to separate the collection of urine and feces at the source for use in agriculture [70]. Other studies have corroborated this idea, suggesting that models that avoid mixing urine with feces would increase the recovery of nutrients. According to Jonathan (2014) [71], urine separation at source, such as the use of urine-separating toilets or the installation of urinals in households and public places, would facilitate this process. But it is important to clarify that in urban areas, the separator toilet can be installed in new buildings and a urine collection cistern could be built in the basement of the building, in addition to the need for a collection truck to take it to a WWTP. Also, the WWTP should have a specific place for the separate treatment of feces from urine. It should be considered that each of these steps generates an additional cost. In rural areas, the installation of the separate toilet is easier, as each house has an individual solution, but cisterns should be built for the storage and further treatment of urine and production of struvite [71].

In urine-separating toilets, urine is piped to collection tanks where a sludge made up largely of struvite ((NH_4_)Mg(PO_4_)·6H_2_O) is formed due to the accumulation of urease. Composed of phosphate, magnesium, and ammonia, struvite has been shown to be an excellent alternative source of essential nutrients for plants. The formation of struvite occurs due to the breakdown of urea into ammonia, causing the pH of the urine to rise, resulting in the precipitation of phosphate, magnesium, calcium, and ammonia. Around 30% of urinary phosphorus is eventually converted to sludge. The concentration of phosphorous in this bottom sludge can be more than twice as high as in the rest of the urine, meaning that it can be used for crops with a high phosphorous demand [69].

Treated wastewater has become a particularly attractive alternative for agriculture in peri-urban and urban areas. It is estimated that 380 billion cubic meters of wastewater are produced annually worldwide, which is equivalent to around 15% of the water abstracted for agriculture. The irrigation potential of this wastewater is 42 million hectares [34].

## 6. Is Energy Recovery from Wastewater Viable in Brazil?

Brazil began to incentivize the production of biofuels in the 1970s with the creation of the National Alcohol Program or *Proálcool*. Three decades later, in 2004, the government launched the National Biodiesel Program [72].

These initiatives contributed to a rapid increase in the share of renewable energy sources in Brazil’s energy matrix. In 2021, 48.3% of the energy produced in the country was renewable, consisting of sugar cane biomass (19.1%), hydro energy (12.6%), firewood and charcoal burning (8.9%), and others (7.7%)—leachate, wind, biodiesel, biogas, solar, and charcoal gas. Brazil is far ahead of other OCDE countries when it comes to the use of renewable energy (Graph 1). While the rest of the world continues to be highly dependent on fossil fuels, Brazil is investing in low-sulfur and aromatic content biodegradable biofuels, which have less impact on the environment than fossil fuels [72].

The most commonly used biofuels in Brazil are ethanol, derived from sugarcane, and biodiesel, produced from vegetable and animal fat. Biodiesel is added to diesel, which is derived from oil, in varying and increasing portions [72].

The production of biodiesel was encouraged by successive laws that provided for the addition of increasing proportions of biodiesel to diesel (Law 11097, which set the limit at between 2% and 5% between 2008 and 2010; Ministerial Order 647, issued in 2014, increasing the limit to 7%; and Law 13263, governing the period 2016 to 2020, which increased the limit further to between 8% and 11%). In response to the COVID-19 pandemic and risk of shortages, the National Petroleum Agency reduced the limit to 10% during some months of 2020 [73] (Figure 3).

Brazil’s biodiesel production capacity increased from 5.35 billion liters in 2018 to 6.4 billion liters in 2020. Production is concentrated in the South (42.6%) and Midwest (39.8%) regions, which are the largest producers of soybeans, the main raw material used in the production of biodiesel [72] (Figure 4).

Recently, the use of algae harvested from wastewater has emerged as a third source of biodiesel. Apart from high algae growth rates and the production of large quantities of biomass (15 to 25 tons per hectare per year), the cultivation of algae for biofuel production uses land that is not suitable for agriculture [75]. Algae can be cultivated in domestic or industrial effluent and capture CO_2_ from both the atmosphere (helping to reduce greenhouse gases) and sewage during growth [76]. Every ton of microalgae used to produce biodiesel can capture up to 2.5 tons of CO_2_ [77,78,79]. Different methods can be used to produce algae, from ponds to photobioreactors [80].

A comparative study of different sources of biodiesel in the United States found that the oil productivity of algae was between 132 and 307 times higher than that of soybeans depending on the type of alga (30% or 70% oil). The soybean oil yield was 446 L/ha (needing 594 million hectares of land area), while the yield of 70% oil microalgae produced in photobioreactors was 136,900 L/ha (needing only 2 million hectares of land area) [81].

This potential has driven the production of biodiesel derived from algae in some countries, including Japan, the United States, Israel, Germany, Portugal, Switzerland, Argentina, and Spain. Brazil has the ideal conditions to produce biodiesel from algae [82].

Brazil also possesses a diverse range of nutrient sources (wastewater by-products such as vinasse, pig waste, ethanol, and palm oil). The federal government, state companies (such as Petrobras and Embrapa), federal universities (in Rio Grande do Sul, Santa Catarina, Paraná, Rio de Janeiro, Bahia, Rio Grande do Norte, and Ceará), and some private companies have played a leading role in promoting research and the development of production, but algae cultivation is still at an experimental stage [83].

A study of prospects for biodiesel production from algae-based wastewater treatment in Brazil [80] analyzed current algae production methods in the country, encompassing each stage of production from cultivation and harvest (including separation of the algae from the liquid growth medium) to oil extraction and conversion into biodiesel. Regarding the extraction of lipids, traditional processes with the use of solvents also extract pigments such as chlorophyll, which, when associated with magnesium, can reduce the quality of the biodiesel produced [84]. There is another method called wet lipid extraction (WLEP), that extracts 60% of the transesterifiable lipids and reduces the need for organic solvents. In this method, chlorophyll is precipitated and reduces chlorophyll contamination in the lipid extract of the algae [84]. At a production rate of 100,000 kg of algae per year, in a raceway pond, 7828 m^2^ of area is required. The production rate under this condition is 0.017 kg/m^3^/d or 0.035 kg/m^2^/d of algae that results in 99.4 m^3^/ha of oil [81]. Algal crude oil has an energy content of 35,800 kJ/kg, around 80% of the energy contained in petroleum [85]. The production of biodiesel from algae cultivated in wastewater has the potential to increase annual biodiesel production by 21%. In addition, based on biodiesel auction prices in February 2015 ($0.728 per liter), revenues from the sale of biodiesel produced from wastewater algae would exceed production costs by 10% [80]. Below, we present an updated version of this perspective.

According to Chisti (2013) [86], 0.35 L of algae oil can be extracted from every 1000 L of wastewater. Based on research by El Shimi et al. (2013) [87], using the transesterification process, 100 L of algae oil can produce 84.7 L of biodiesel. Rounding this quantity down to 80 L of biodiesel for every 100 L of algae oil, 1000 L of wastewater has the potential to produce 0.28 L of biodiesel.

Based on the total volume of wastewater collected in the country (6.0 × 10^12^ L), 1.68 × 10^9^ of biodiesel could have been produced in Brazil in 2020 [88]. Given that the total biodiesel production in 2020 was 6.432 × 10^9^ L, the use of domestic wastewater would represent an increase in production of 26.11%. In 2020, Brazil would have had the potential to produce 8.11 × 10^9^ L of biodiesel, ahead of the United States, which produced 6.88 × 10^9^ L.

The 82nd auction of biodiesel held by the National Petroleum Agency in August 2021 set a price of $1.04 per liter (the exchange rate on 31 October 2022 was $1 to R$5.65 (Brazilian Real)). Figure 5 shows trends in the mean annual prices obtained in biodiesel auctions between 2010 and 2021 [74].

A more ambitious calculation could be performed based on the assumption of universal access to sanitation services in the country. Considering that daily per capita water consumption in Brazil in 2020 was 152.1 L [88] and that wastewater generation is estimated to be 80% of water consumption, daily per capita wastewater production was approximately 122 L (or 4.53 × 10^4^ L per capita over the year), which is equivalent to 12.47 L (multiplying 122 L by 365 days (1 year) resulting in 4.53 × 10^4^ L of wastewater over the year, multiplied by 0.28 L (El Shimi et al., 2013) [77] giving the amount of biodiesel produced per person in 2020 (12.47 L)) of biodiesel per capita.

The cost of producing biodiesel from algae cultivated in wastewater is $3.90 per liter [89]. Considering that the price per liter of biodiesel in the 82nd Brazilian auction was $1.04, the net cost of the production of biodiesel will reduce to $2.86 per liter, or 26.7%. However, if we consider the value-added products (VAPs), such as protein, astaxanthin, and other products, such as cosmetics from algae, the cost of biodiesel production could be reduced to $0.54 per liter [89]. With the price that was given in the 82nd auction, $1.04 per liter, the cost of production will be turned to a profit of $0.50 per liter, an amount that could be used to lower the cost of wastewater treatment.

This can be viewed as a major incentive for investing in wastewater treatment. In addition to the political, public health, and quality of life gains produced by achieving universal access to basic sanitation, the conversion of wastewater into biofuel provides significant economic benefits, allowing Brazil to increase biodiesel production by 26.11% (reaching 8.11 × 10^9^ L), moving ahead of the United States, which produced 6.88 × 10^9^ L in 2020. The production of biodiesel from algae-based wastewater treatment transforms a problem—water pollution generated by wastewater—into a solution with multiple gains [72]. However, there are some issues to be considered [86]. The cost of production is high and becomes viable as the production increases. The production of 1 ton of algae requires 1.83 tons of carbon dioxide, and it represents around 1/3 of the cost of producing biomass. The production also requires nitrogen and phosphorus, and the fertilizers used in agriculture are insufficient for large-scale production. Some alternatives would be the use of cyanobacteria that fix nitrogen and the use of biogas from anaerobic digestion. The energy required for pumping and filtration could be obtained from fossil fuels. The biodiesel to be produced through this process would contain some amount of energy that should be at least seven times the energy required for its production. To reach this threshold, an alternative is to use the biomass residue to generate biogas to be used as fuel to run the production. It is also necessary to analyze if the carbon footprint, which is the amount of carbon released by the use of biodiesel, is lower than that of petroleum [86]. Finally, the price of oil fluctuates, and this influences the price of biodiesel. Nevertheless, algae have a promising market. It is estimated that the market for the value of algae-derived products (VAPs) will reach $53.43 billion in 2026. The VAPs considered are vitamins, terpenoid, flavonoids, pigments, alginate, agar, protein, astaxanthrin, hormones, etc. Algae have applications in the health sector as anticancer, anti-inflammatory, antimicrobial, anticoagulating, and cholesterol-lowering agents. In terms of sewage treatment, algae have achieved a greater than 90% removal of nutrients and organic load, and this contributes to increasing the methane concentration in biogas [89].

## 7. Conclusions

The aim of this article is to show the path toward sustainability in wastewater management and what stage Brazil is at. It shows that Brazil is still at stage 1 (prevention of or reduction in pollution at the source), although it has already achieved some items related to stage 2 (wastewater collection and treatment). However, some promising initiatives are described in this article that are associated with stage 3 (using wastewater as an alternative source of water) and stage 4 (use of wastewater by-products). Regarding stage 3, there is no legislation governing water recycling and the reuse of wastewater is still in the early stages of development in the country. Concerning stage 4, while Brazil has begun to implement the production of biogas in some WWTPs, concrete efforts are required to harness the potential of nitrogen and phosphorous in urine and feces, including the promotion of the use of urine-separating toilets. Another promising option is the cultivation of algae in wastewater to produce biodiesel, increasing the volume of biodiesel produced by the country. Brazil, a country in which 48.3% of energy production comes from renewable sources, can adopt a win-win solution with double gains: universal access to basic sanitation and an increase in biodiesel production. However, the biodiesel market rate is subject to fluctuations, impacting the profitability of the production of biodiesel from algae-based wastewater treatment. Nonetheless, this alternative could reduce the cost of wastewater treatment, as the profit from the production of biodiesel considering the value-added products from algae (VAPs) is around $0.50 per liter. The VAPs consist of cosmetics, medicines, etc. This promising situation will only be possible if there is political will and investment from federal, state, and municipal governments in wastewater collection and treatment infrastructure.

## Figures and Tables

**Figure 1 ijerph-20-06597-f001:**
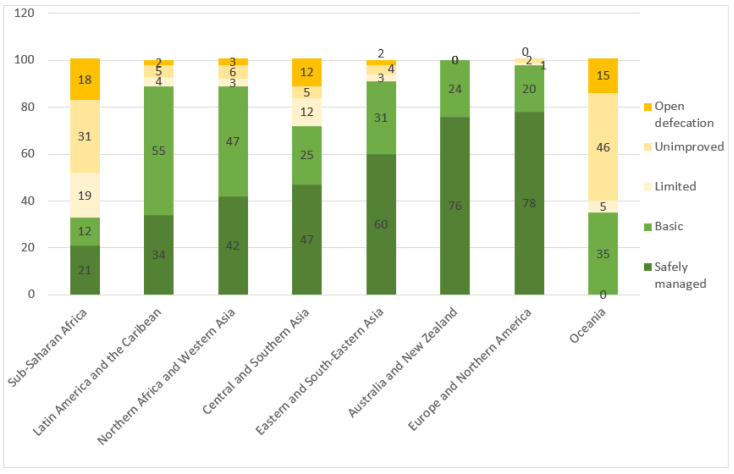
Percentage of sanitation coverage by service level and region (2020) (%). Source: UNICEF & WHO, 2021 [8].

**Figure 2 ijerph-20-06597-f002:**
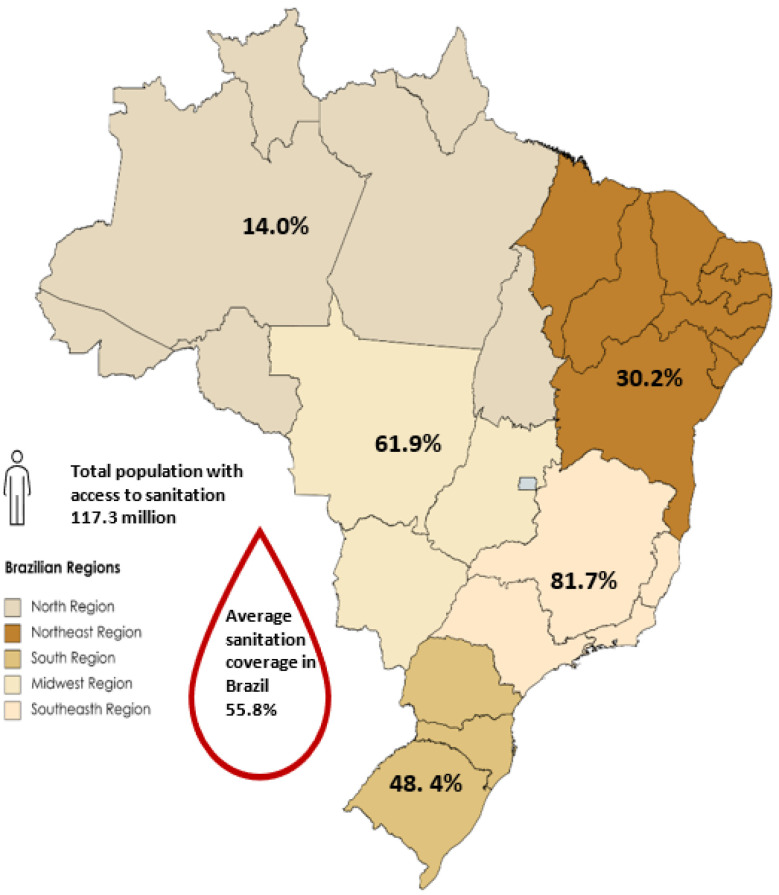
Sanitation coverage in Brazil (2022). Source: BRASIL, 2022 [13].

**Figure 3 ijerph-20-06597-f003:**
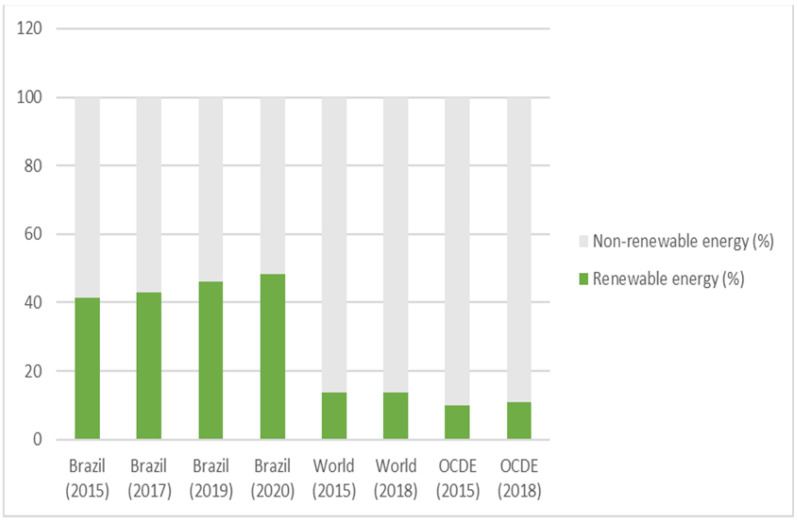
Share of renewable energy in the energy matrix. Source: EPE, 2021 [73].

**Figure 4 ijerph-20-06597-f004:**
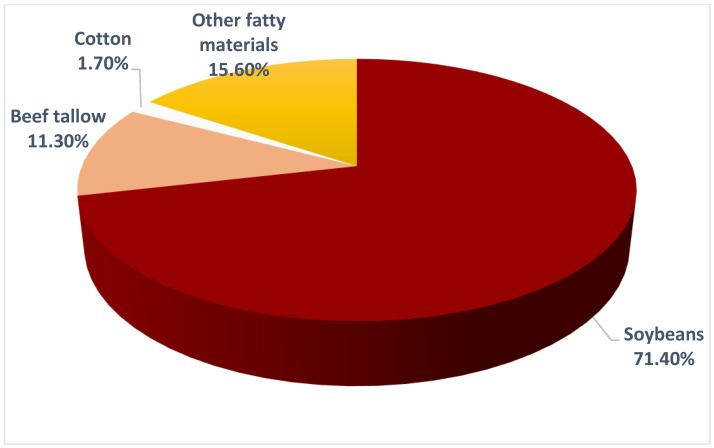
Composition of biodiesel in Brazil. Source: ANP, 2021 [74].

**Figure 5 ijerph-20-06597-f005:**
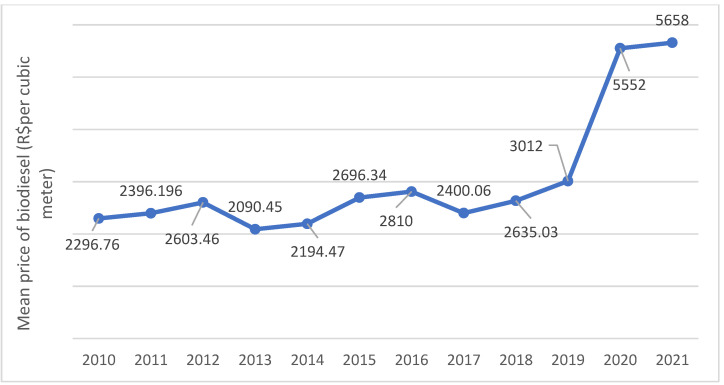
Trends in biodiesel auction prices in Brazil. Source: ANP, 2021 [74].

**Table 1 ijerph-20-06597-t001:** Ladder for sanitation services. Source: adapted from UNICEF & WHO, 2017 [7].

Service Level	Definition
Safe management	Each household has its own facility on its own land where excreta is disposed on site or transported to a sewage treatment plant
Basic	Each family has its own facility on its own land
Limited	The sanitary facility is used by two or more families
Unimproved	Use of improvised latrines
Open defecation	Not using toilets to defecate

## Data Availability

The data consulted to write this article were from a literature review on sites such as Web of Science, PubMed, and others and also data from the National Sanitation Information System (SNIS), obtained from the Secretary of Sanitation of the Ministry of Regional Development, in addition to WASH reports obtained from the UNICEF and WHO sites and also a report on sewage management obtained from UNESCO.

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
