# Peer review of "Path toward Sustainability in Wastewater Management in Brazil"

_ijerph, 2023, doi:10.3390/ijerph20166597_

Round 1

Reviewer 1 Report

This article focuses on sustainability in wastewater management and the use of wastewater as a raw material for energy production as a path to sustainability in Brazil. However, the article also discusses other aspects of sustainability in wastewater management related to the circular economy, such as reuse of wastewater and recovery of nutrients and other products.

In general, the article is well written and the topic is important when considering the situation of wastewater treatment in developing countries. The whole discussion is built over a UNESCO document where wastewater management is divided into four "stages", from basic wastewater treatment to bioproduct recovery.

I think the article is appropriate for publication in the journal. However, I have some comments that might help improve the article:

1. I know this is a content issue, but it looks like the article uses "sustainable development" and "sustainability" as synonyms. Personally, I think Brundtland's concept is very old (more than 30 years). One could have a long discussion on this point, but that is unnecessary in this article. I would only suggest that the textual use of the concept of sustainable development is no longer necessary and that the term "sustainability" instead "sustainable development" be used in this article.

2. Line 116: I would think that many "underdeveloped" countries already have regulations on this issue, including Argentina, Chile, and Uruguay, as examples in the same region. However, this does not necessarily guarantee that regulatory instruments are properly implemented or that management is adequate.

3. Line 129: There is a very extensive discussion on this topic, and it is difficult to argue that onsite wastewater is more appropriate than centralized treatment if many factors are not considered, such as residential density, soil  and groundwater characteristics, among other factors. You can see these articles only as an example and introduction to the topic:
Bernal et al. (2021). DOI: 10.1016/j.heliyon.2021.e06375
Kalbar and Lokhande (2023). DOI: 10.2166/wp.2023.267

4. In lines 160 to 223, the authors describe the situation in Brazil, but do not even mention the potential pollution problems of groundwater related by onsite sewage, which is a consequence of poor management of septic systems.

5. The conclusion is very brief and does not include a deeper discussion of the policy dimension.

Author Response

Response to Reviewer 1 Comments

Point 1: I know this is a content issue, but it looks like the article uses "sustainable development" and "sustainability" as synonyms. Personally, I think Brundtland's concept is very old (more than 30 years). One could have a long discussion on this point, but that is unnecessary in this article. I would only suggest that the textual use of the concept of sustainable development is no longer necessary and that the term "sustainability" instead "sustainable development" be used in this article.

Response 1: We substitute sustainable development for sustainability

Point 2: Line 116: I would think that many "underdeveloped" countries already have regulations on this issue, including Argentina, Chile, and Uruguay, as examples in the same region. However, this does not necessarily guarantee that regulatory instruments are properly implemented or that management is adequate.

Response 2: We insert the word “Many” with “countries with regulatory instruments and enforcement processes in place have completed this stage and at the end of paragraph we include the frase “However, this does not necessarily guarantee that regulatory instruments are properly implemented or that management is adequate. (Now this is in line 126/127)

Point 3: Line 129: There is a very extensive discussion on this topic, and it is difficult to argue that onsite wastewater is more appropriate than centralized treatment if many factors are not considered, such as residential density, soil  and groundwater characteristics, among other factors. You can see these articles only as an example and introduction to the topic:
Bernal et al. (2021). DOI: 10.1016/j.heliyon.2021.e06375
Kalbar and Lokhande (2023). DOI: 10.2166/wp.2023.267

Response 3: We do not judge whether centralized or decentralized treatment is right. The reference that we have consulted, have observed that developed countries treat 70%, developing countries 28% to 38% and the poorest countries only 8%. Usually what happens are centralized solutions, but there are countries where it is possible to use decentralized solutions, which has the advantage of being cheaper. Also in the reference, Bernal et al, 2021, there is the same consideration that we have exposed here, in the paragraph:

““With respect to wastewater, its treatment and disposal is widely practiced by developing countries, with a high percentage of the population (over 90%) connected to centralized treatment systems (Burton et al., 2014). However, decentralized systems are becoming particularly attractive because of the possibility of reducing long-term treatment costs and the potential for wastewater reuse (Daigger, 2009; Roefs et al., 2017; Jung et al., 2018).”

Point 4. In lines 160 to 223, the authors describe the situation in Brazil, but do not even mention the potential pollution problems of groundwater related by onsite sewage, which is a consequence of poor management of septic systems.

Response 4: Now the line is 212, we include "and groundwater" and a paragraph, with another bibliographic reference:

VILLAR,P.C., HIRATA,R.,SUBOGUSOFF,A.V, MARCELLINI,S.S., MARCELLINI,L. As águas subterrâneas e o saneamento no Brasil: importância e desafios. In: SILVA,J.I.A.O.(org.), O problema da água e do saneamento : algumas respostas, volume 1, 1ª edição, Campina Grande:Editora da Universidade Estadual da Paraíba (EDUEPB), 2021, parte 3, capitulo1, p.115-132. Disponível em: https://www.researchgate.net/publication/356972461_O_problema_da_agua_e_do_saneamento_algumas_respostas_Volume_1_O_problema_da_agua_e_do_saneamento_algumas_respostas_Volume_1. Acesso:13/07/2023

Point 5. The conclusion is very brief and does not include a deeper discussion of the policy dimension.

Response 5: I have expanded the conclusion and put in the political aspect.

Reviewer 2 Report

The manuscript deals with an interesting subject, although some aspects need corrections. In general, it is a well-written document easy to understand and follow. However, authors should set their point regarding a practical solution to reaching sustainability goals. Comments regarding the present stage of technology and associated energy demand are needed. Otherwise, this document just becomes a list of desires, but no helpful information can be extracted from it since technical approaches are not given. Therefore, comments regarding the feasibility of implementing technological solutions should be introduced given the population distribution and type of living in Brazil and considering people's incomes. 

1. please check the reference position. Space is missing.

2. P1 L46 GDP needs definition

3. P4 L135-159: Please add comments (using suitable references) regarding the risks of using recycled wastewater. Water treatment systems are costly and many countries lack adequate sanitary services because installation and maintenance costs are very high. Income distribution is not regular in the different population statuses. Add comments regarding poverty in these countries, creating added difficulties in keeping and maintaining sanitary services.

4. What currency is R$? Please use dollars or euros to express currency, so a wider audience will better understand currency magnitudes.

5. L216-221: correct font size

6. Add comments regarding the different income in the south and northern regions of Brazil, the way of living of poor people and house distribution. Illegal settlements are also reasons for not providing sanitary services. A solution needs first to be provided to reduce poverty in these regions.

7. L250-253: Add comments regarding detergents and soap types for the safe use of greywater to avoid soil compaction and plant poisoning. These products usually have higher prices. Therefore, in zones with low incomes, the probability of people using expensive products will be low, thus generating additional problems. The use of greywater also means a new sewage installation, so these are solutions for new settlements but not for existing ones.

8. L278-295: Please add comments regarding the difficulties in implementing these measurements in poor settlements. The document, at this point, is just a list of desires but does not address difficulties found in implementing measurement. An accurate approximation of the problem will aid in finding a solution. Add photographs showing how are the locations where not sanitary services are available. And indicate how the authors think solutions should be provided.

9. The content of section 4 is not in accordance with the title. It is just a list of desires and generalities but does not focus on real possibilities and providing feasible solutions to the problem. Add energy demand and costs for treating wastewater. The size of the plant, values of levelized costs and alternatives based on population density, income, city ground, and terrain orography. Circular economy proposes the reuse and recycling of materials, but this is achieved by increasing the energy demand of cycles. This is a thermodynamic rule which cannot be broken. The use of digested sludge as fertilizer is full of restrictions, and European regulation is not clear in many cases. Thus digestate is more of a problem for many WWTP than a valuable product. How are authors supposed to achieve a solution in Brazil when there are so many aspects still waiting for a solution in developed countries?

10. L390-397: Please indicate that these findings are just experimental work and far from industrial applications. Brazil and Latin America need real solutions which can be implemented in the short term with technology that is feasible and reliable. 

11. L399-404: Add comments regarding WWTP energy demand. Electricity production by digestion of sewage sludge scarcely covers 60% of the total demand, so the proposal of the authors is not really a solution to energy problems. Digestion just helps in reducing the high energy demand for treating wastewater. The fact is that developing countries find it hard to cover the costs of new WWTP and pay the energy demand of these units. Comments regarding these aspects should be added using suitable references regarding the energy demand per cubic meter of wastewater.

12. L436-440: Give a clue of how the separation of urine and feces should be implemented in new housing and how this should be performed in existing ones. Probably this idea may only work for new house constructions. Please also indicate that agricultural land is usually far from the WWTP location so the spreading of digestate needs to take into account the transport of the material. Sludge drying to reduce transport also sets a burden on final disposal because of transport costs. 

13. L486-493: Indicate the experimental stage of these studies. Please add comments regarding the problems of extracting lipids due to the presence of chlorophyll.

14. L517-526: Add data regarding the surface area needed for microalgae cultivation, productivity, and energy demand of algae ponds if the circulation of water is needed. 

15. L529, L538, L542: correct superscript.

16. Please correct the currency of the whole document to US dollars or euros, or add currency change to these quantities

17.  L533-538. This is a simplistic approach since the installation and maintenance costs make this alternative not profitable. WWTPs do not generate enough resources to offset the energy demand. Thus, the installation of WWTP for treating all septic water will raise the country's energy demand instead of providing an additional energy source. Add suitable references to make comments regarding this point

18. L542-545: Check the costs of producing biodiesel from microalgae. Dehydration, drying of biomass, and extraction of lipids are costly stages that reduce profit. Check references regarding biodiesel obtention from biomass microalgae.

19. If accounting were so straightforward and benefits so clear, sure the problem would have already been solved. Please indicate all complexities found and difficulties in attempting the aim. Approximating the real problem is a straightforward way to find a solution. The document indicates desires but denies difficulties and problems of the approach. Thus becoming a compendium of wishes. Please cover all aspects of the analysis so this document will help understand difficulties in attaining sustainable goals. So adequate resources can be provided and aid can be specific to attain a solution  

20. L557: Authors do not show. They cite other documents regarding this point. So the correct phrase should indicate that the country is in stage 1.

21. L563-565: Indicate if this alternative is being proposed for new construction settlements or are authors proposing this for existing city sewage installation? 

22. Indicate that algae cultivation is at an experimental stage, so this is not a practical solution currently with well-developed and profitable technology available.

Author Response

Response to Reviewer 2 Comments

Point 1: please check the reference position. Space is missing.

Response 1: We check the reference position.

Point 2: P1 L46 GDP needs definition.

Response 2: we insert the definition of GDP (now is line 47/48)

Point 3: P4 L135-159: Please add comments (using suitable references) regarding the risks of using recycled wastewater. Water treatment systems are costly, and many countries lack adequate sanitary services because installation and maintenance costs are very high. Income distribution is not regular in the different population statuses. Add comments regarding poverty in these countries, creating added difficulties in keeping and maintaining sanitary services.

Response 3: Risk and the cost of treatment was added in paragraph (L154-164) and a new reference was used.

Duong,K., Saphores, J-D.M. Obstacles to wastewater reuse: an overview. WIREs Water 2015, 2:199–214. doi: 10.1002/wat2.1074

Point 4. What currency is R$? Please use dollars or euros to express currency, so a wider audience will better understand currency magnitudes.

Response 4: we substitute R$ (Brazilian money) for dolars

Point 5. L216-221: correct font size

Response 5: we correct the font size.

Point 6. Add comments regarding the different income in the south and northern regions of Brazil, the way of living of poor people and house distribution. Illegal settlements are also reasons for not providing sanitary services. A solution needs first to be provided to reduce poverty in these regions.

Response 6: We insert a paragraph with the variation of sanitation index and percentage of poor population (from L201 – 208)

Point 7.  L250-253: Add comments regarding detergents and soap types for the safe use of greywater to avoid soil compaction and plant poisoning. These products usually have higher prices. Therefore, in zones with low incomes, the probability of people using expensive products will be low, thus generating additional problems. The use of greywater also means a new sewage installation, so these are solutions for new settlements but not for existing ones.

Response 7: It was included a paragraph (L 293-295) and a new reference.

Arun K. Vuppaladadiyam . Noemi Merayo . Pepijn Prinsen . Rafael Luque . Angeles Blanco . Ming Zhao, A review on greywater reuse: quality, risks, barriers and global scenarios, Rev Environ Sci Biotechnol (2019) 18:77–99 . https://doi.org/10.1007/s11157-018-9487-9

Point 8. L278-295: Please add comments regarding the difficulties in implementing these measurements in poor settlements. The document, at this point, is just a list of desires but does not address difficulties found in implementing measurement. An accurate approximation of the problem will aid in finding a solution. Add photographs showing how are the locations where not sanitary services are available. And indicate how the authors think solutions should be provided.

Response 8: we insert a paragraph (L336-341) including what the reviewer asked. We use 3 new references:

Reference 1 – Vuppaladadiyam, A.K., Merayo, N.,Prinsen, P., Luque, R., Blanco,A., Zhao,M., A review on greywater reuse: quality, risks, barriers and global scenarios, Rev Environ Sci Biotechnol (2019) 18:77–99, https://doi.org/10.1007/s11157-018-9487-9(0123

Reference 2 – Porto, M. F. S., Cunha, M.B., Pivetta, F., Zancan, L., Freitas, J.D., Saúde e ambiente na favela: reflexões para uma promoção emancipatória da saúde. Serv. Soc. Soc. (123) • Jul-Sep 2015 • https://doi.org/10.1590/0101-6628.035 

Referência 3 – Katukiza, A.Y., Ronteltap, M., Niwagaba, C.B., Kansiim, F., Lens, P.N.L., Grey water treatment in urban slums by a filtration system: Optimisation of the filtration médium. Journal of Environmental Management 146 (2014) 131-141. (http://dx.doi.org/10.1016/j.jenvman.2014.07.033

Point 9. The content of section 4 is not in accordance with the title. It is just a list of desires and generalities but does not focus on real possibilities and providing feasible solutions to the problem. Add energy demand and costs for treating wastewater. The size of the plant, values of levelized costs and alternatives based on population density, income, city ground, and terrain orography. Circular economy proposes the reuse and recycling of materials, but this is achieved by increasing the energy demand of cycles. This is a thermodynamic rule which cannot be broken. The use of digested sludge as fertilizer is full of restrictions, and European regulation is not clear in many cases. Thus digestate is more of a problem for many WWTP than a valuable product. How are authors supposed to achieve a solution in Brazil when there are so many aspects still waiting for a solution in developed countries?

Response 9: We changed the numbering from 4 to 5 and included the other two items as subtopics, they became 5.1 (biogas) and 5.2(Agricultural Use of Sewage Sludge) because they are by By- products of wasteater (item 5). We include a paragraph (L-464-470) with values of energy demand from Anaerobic Digestion. We include a new reference:

Ranieri,E., Giuliano,S., Ranieri, A.C. Energy consumption in anaerobic and aerobic based wastewater treatment plants in Italy, Water Practice & Technology Vol 16 No 3, 2021

doi: 10.2166/wpt.2021.045

Point 10. L390-397: Please indicate that these findings are just experimental work and far from industrial applications. Brazil and Latin America need real solutions which can be implemented in the short term with technology that is feasible and reliable.

Response 10: We have taken the reviewer's suggestion and included a sentence in the text (L-474-476).

Point 11. L399-404: Add comments regarding WWTP energy demand. Electricity production by digestion of sewage sludge scarcely covers 60% of the total demand, so the proposal of the authors is not really a solution to energy problems. Digestion just helps in reducing the high energy demand for treating wastewater. The fact is that developing countries find it hard to cover the costs of new WWTP and pay the energy demand of these units. Comments regarding these aspects should be added using suitable references regarding the energy demand per cubic meter of wastewater.

Response 11: We have added a paragraph (L482-489) and 2 new bibliographic references. We explain that the balance between energy production by anaerobic digestion and energy consumption is not favorable but that there is a reduction in electricity costs.

Reference 1 – Bilotta, P., Ross,B.Z.L., Estimativa de geração de energia e emissão evitada de gás de efeito estufa na recuperação de biogás produzido em estação de tratamento de esgotos. Eng. Sanit. Ambient. 21 (2) • Apr-Jun 2016 •https://doi.org/10.1590/S1413-41522016141477

Reference 2 – Maktabifard, M., Zaborowska,E., Makinia,J. Achieving energy neutrality in wastewater treatment plants through energy savings and enhancing renewable energy production. Rev Environ Sci Biotechnol (2018) 17:655–689 https://doi.org/10.1007/s11157-018-9478-x

Point 12. L436-440: Give a clue of how the separation of urine and feces should be implemented in new housing and how this should be performed in existing ones. Probably this idea may only work for new house constructions. Please also indicate that agricultural land is usually far from the WWTP location so the spreading of digestate needs to take into account the transport of the material. Sludge drying to reduce transport also sets a burden on final disposal because of transport costs. 

Response 12: We have already written in the text (L 529-536) explaining that in the urban area only in new construction but, it will be necessary to transport truck and this will incur additional cost. In the rural area, since there is an individualized solution, it can be implemented.

Point 13. L486-493: Indicate the experimental stage of these studies. Please add comments regarding the problems of extracting lipids due to the presence of chlorophyll.

Response 13: We add a line in line 604, and a paragraph (L 609-618). We add a reference:

Sathish,A., Sims, R.C., Biodiesel from mixed cultures algae via a wet lipid extraction procedure. Bioresource tecnology, vol. 118, August 2012, p.643-647. https://doi.org/10.1016/j.biortech.2012.05.118

Chisti, Y. Biodiesel from microalgae.Biotechnology Advances 25 (2007) 294–306

Chisti, Y., 2012. Raceways-based production of algal crude oil. In: Posten, C., Walter,C. (Eds.), Microalgal Biotechnology: Potential and Production. de Gruyter, Berlin,pp. 113–146.

Point 14. L517-526: Add data regarding the surface area needed for microalgae cultivation, productivity, and energy demand of algae ponds if the circulation of water is needed

Response 14: We have inserted a sentence in a previous paragraph (L-609-617), so as not to change the meaning of the text by including the information the reviewer requested.

Point 15. L529, L538, L542: correct superscript

Response 15: We correct.

Point 16. Please correct the currency of the whole document to US dollars or euros, or add currency change to these quantities

Response 16: We correct.

Point 17. This is a simplistic approach since the installation and maintenance costs make this alternative not profitable. WWTPs do not generate enough resources to offset the energy demand. Thus, the installation of WWTP for treating all septic water will raise the country's energy demand instead of providing an additional energy source. Add suitable references to make comments regarding this point

Response 17: The bibliographic reference shown earlier in the question on energy production by anaerobic digestion and electricity demand (point 11), showed that several countries are optimizing energy production by anaerobic digestion to reach neutrality, i.e., reduce energy consumption in sewage treatment. We add a paragraph (L482-489)

Reference: Maktabifard, M., Zaborowska,E., Makinia,J. Achieving energy neutrality in wastewater treatment plants through energy savings and enhancing renewable energy production. Rev Environ Sci Biotechnol (2018) 17:655–689 https://doi.org/10.1007/s11157-018-9478-x

Point 18: L542-545: Check the costs of producing biodiesel from microalgae. Dehydration, drying of biomass, and extraction of lipids are costly stages that reduce profit. Check references regarding biodiesel obtention from biomass microalgae

Response 18: We have changed the cost calculation with more up-to-date data (L645- 652) and used 2 new references.
Rafa, N., Ahmed, S.F.,Badruddin, I.A., Mofijur, M., Kamangar, S., Strategies to Produce Cost-Effective Third- Generation Biofuel from Microalgae, Frontiers in Energy Research, September 2021, volume 9. Volume 9, doi: 10.3389/fenrg.2021.749968

Branco-Vieira, M., Mata, T.M., Martins, A.A., Freitas, M.A.V., Caetano, N.S. Economic analysis of microalgae biodiesel production in a small-scale facility. Energy reports, vol.6, supplement 8, December 2020, p. 325-332.

Point 19: If accounting were so straightforward and benefits so clear, sure the problem would have already been solved. Please indicate all complexities found and difficulties in attempting the aim. Approximating the real problem is a straightforward way to find a solution. The document indicates desires but denies difficulties and problems of the approach. Thus becoming a compendium of wishes. Please cover all aspects of the analysis so this document will help understand difficulties in attaining sustainable goals. So adequate resources can be provided and aid can be specific to attain a solution

Response 19: We have inserted a paragraph before the conclusion (L-631-653) explaining the problems that must be overcome. (L667-686)

In the paragraph indicated there is only an estimate of given the water consumption, 80% becomes sewage and from the estimated volume of sewage produced in Brazil, estimate the production of biodiesel and by the last auction the price to be acquired. It is just an exercise to show that according to the price of biodiesel in the auctions, the production of biodiesel from sewage can be profitable and cover the costs of sewage collection and treatment. Nothing was said about demand or energy consumption in WWTP. In the paragraph indicated there is only the estimation of sewage production through water consumption and how much this would represent in terms of biodiesel production.

Point 20: L557: Authors do not show. They cite other documents regarding this point. So the correct phrase should indicate that the country is in stage 1.

Response 20: we have corrected in the conclusion.

Point 21: L563-565: Indicate if this alternative is being proposed for new construction settlements or are authors proposing this for existing city sewage installation? 

Response 21: The authors are proposing for new construction or to be used in rural areas or in area with individualized solutions.

Point 22: Indicate that algae cultivation is at an experimental stage, so this is not a practical solution currently with well-developed and profitable technology available.

Response 22: We inserted a sentence saying that algae cultivation is in the experimental phase (L-604)

Round 2

Reviewer 1 Report

The paper have been improved. I think can be published at the present form.

Author Response

Our group appreciates your review.

Reviewer 2 Report

The manuscript has been greatly improved; minor details are given below.

L599: XOCs needs to be defined

L714, L843: English edition

L994: please check if unit PE has been previously defined

L999: Subscript

L1106-1112: use always the same format for expressing units, m3/d and kWh/m3

ídem for L1383-1392, check the use of kJ, kg, lowercase letter for "k"

L1408: What is 82 st? U$ is US$?, please clarify and make sure in the document is clear

L1487: What is the difference between U$ and $ in this paragraph. Please check currency is clear in the whole document.

Please check English

Author Response

Response to Reviewer 1 (Round 2) Comments

Our group appreciates your review and corrections.

Response to Reviewer 2 (Round 2) Comments

Point 1: L599: XOCs needs to be defined.

Response 1: Xenobiotic organic compounds, we dropped the abbreviation, spelled out the meaning and changed the text.

Point 2: L714, L843: English edition. (348?)

Response 2: We had difficulty finding the marked line after editing the text, but we revised the English.

Point 3: L994: please check if unit PE has been previously defined.

Response 3: ok. We correct.

Point 4. L999: Subscript

Response 4: ok. We correct.

Point 5. L1106-1112: use always the same format for expressing units, m3/d and kWh/m3

Response 5: ok. We correct.

Point 6. ídem for L1383-1392, check the use of kJ, kg, lowercase letter for "k"

Response 6: ok. We correct.

Point 7.  L1408: What is 82 st? U$ is US$?, please clarify and make sure in the document is clear

Response 7: ok. We correct.

Point 8. L1487: What is the difference between U$ and $ in this paragraph. Please check currency is clear in the whole document.

Response 8: ok. We correct.